# Privacy-Preserving Operating Room Workflow Analysis using Digital Twins

**Alejandra Perez**[1]                                                    APEREZR6@JH.EDU
**Han Zhang**[1]
**Yu-Chun Ku**[1]
**Lalithkumar Seenivasan**[1]
**Roger D. Soberanis-Mukul**[1]
[1] *Johns Hopkins University, Baltimore, MD, 21211, USA*

**Jose L. Porras**[2]
**Richard Day**[2]
**Jeff Jopling**[2]
**Peter Najjar**[2]
[2] *Johns Hopkins Medical Institutions, Baltimore, MD, 21287, USA*

**Mathias Unberath**[1]                                                  UNBERATH@JHU.EDU

**Editors:** Accepted for publication at MIDL 2025

## Abstract

The operating room (OR) is a complex environment where optimizing workflows is critical to reduce costs and improve patient outcomes. While computer vision approaches for automatic recognition of perioperative events can identify bottlenecks for OR optimization, privacy concerns limit the use of OR videos for automated event detection. We propose a two-stage pipeline for privacy-preserving OR video analysis and event detection. First, we leverage vision foundation models for depth estimation and semantic segmentation to generate de-identified Digital Twins (DT) of the OR from conventional RGB videos. Second, we employ the SafeOR model, a fused two-stream approach that processes segmentation masks and depth maps for OR event detection. Evaluation on an internal dataset of 38 simulated surgical trials with five event classes shows that our DT-based approach achieves performance on par with—and sometimes better than—raw RGB video-based models for OR event detection. Digital Twins enable privacy-preserving OR workflow analysis, facilitating the sharing of de-identified data across institutions and potentially enhancing model generalizability by mitigating domain-specific appearance differences.

**Keywords:** Operating Room, Workflow Analysis, Digital Twins, Workflow Optimization

## 1. Introduction

The operating room (OR) is a high-stakes environment where optimizing workflows is critical to reduce costs and improve patient outcomes. Standardized surgical processes enhance efficiency by minimizing variability, preventing delays, and maximizing resource utilization (Lee et al., 2019; Nundy et al., 2008). Systematic monitoring of perioperative events enables surgical teams to identify bottlenecks and optimize OR workflows. However, manual reporting of events is labor-intensive and error-prone, highlighting the need for automated solutions. Advances in computer vision now enable the recognition of events and phases from OR videos, offering significant efficiency gains (Sharghi et al., 2020; Özsoy et al., 2024).

Despite these advances, privacy remains a major challenge, as OR videos contain sensitive information about patients, staff, and the environment. Current privacy-preserving approaches such as face blurring (Bastian et al., 2023) or the use of depth images enabled by structured light cameras (Jamal and Mohareri, 2022) have critical limitations. Face blurring only addresses facial features but not other potentially identifiable image content, and – while depth image analysis removes all visual features – structured light cameras require specialized hardware making it impractical for immediate use. To address these limitations, we propose a two-stage pipeline for privacy-preserving OR video analysis and event detection (Fig. 1) that operates directly on RGB video of conventional cameras. In the first stage, we leverage vision foundation models (Yang et al., 2024; Kirillov et al., 2023) for depth estimation and semantic segmentation to generate Digital Twins (DT) of the OR. Since these DT only contain depth and semantic information, they are fully de-identified, while providing a geometric abstraction of the scene (Ding et al., 2024a). The second stage utilizes an independent model to analyze these DT and identify key OR workflow events. Our results indicate that this approach achieves performance on a par, and sometimes even better, than RGB video-based models.

## 2. Method

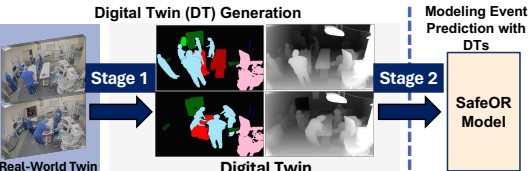
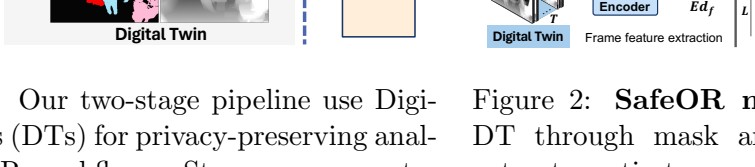

Figure 1: Our two-stage pipeline use Digital Twins (DTs) for privacy-preserving analysis of OR workflows. Stage one generates DTs and Stage two is an independent model trained on these DTs to detect OR events.

Figure 2: **SafeOR model:** Processes the DT through mask and depth streams to extract spatio-temporal features, and uses cross-attention for bi-modal fusion to predict events.

**Stage 1: Digital Twin generation.** A DT is a digital representation of a real-world environment. We leverage segmentation and depth vision foundation models to create 2D DTs of the OR scene. We fine-tune an object detection model (Zong et al., 2023) for bounding boxes that serve as prompts for SAM (Kirillov et al., 2023) to produce segmentation masks scene OR objects, and retrieve monocular depth estimation using the Depth Anything v2 Model (Yang et al., 2024). **Stage 2: SafeOR model for event detection.** We employ a two-stream approach, processing segmentation masks and depth maps independently. As shown in Fig. 2.A, we extract spatio-temporal features by constructing a temporal window around each frame of the video and pass it through a video backbone in each stream. Formally, given a sequence of segmentation masks $\mathcal{M} = \{m_1, \ldots, m_T\}$ and depth maps $\mathcal{D} = \{d_1, \ldots, d_T\}$, where T is the window size, we use separate video encoders to generate mask $\mathbf{E}m_f$ and depth embeddings $\mathbf{E}d_f$, respectively. We then perform cross-attention between a set of depth-stream and mask-stream embeddings of dimension $L$ to

Table 1: **Results on the HALO dataset.**
We report mAP at different tIoU thresholds.

| Input | mAP@0.25 | mAP@0.50 | mAP@0.75 | Avg. mAP |
|---|---|---|---|---|
| RGB | 93.52 | 75.64 | 43.09 | 70.75 |
| Mask DT | **94.84** | 74.30 | 42.07 | 70.40 |
| Depth DT | 90.43 | 77.07 | 37.55 | 68.35 |
| Mask-Depth DT | 92.44 | **79.46** | **46.89** | **72.93** |

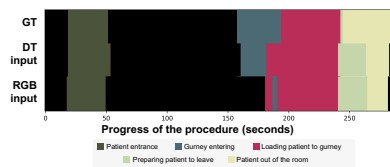

Figure 3: Qualitative results of OR event detection of an example procedure.

Table 2: **Prediction Error (seconds) on HALO dataset.** Per class error of the predicted start and end time of events compared to the ground truth. We compare our Mask-Depth DT against raw RGB input.

| | Mask-Depth DT | | RGB | |
|---|---|---|---|---|
| | Start | End | Start | End |
| Patient Preparation | **5.89 ± 11.51** | **2.63 ±3.74** | 7.89 ± 12.33 | 14.00 ± 39.46 |
| Gurney Entering | **11.65 ±25.15** | **18.20 ± 26.52** | 33.09 ± 75.94 | 38.09 ± 68.96 |
| Loading patient to gurney | 32.62 ± 66.69 | 30.57 ± 75.03 | **27.40 ± 63.55** | **18.25 ± 52.17** |
| Preparing patient to leave | 2.33 ± 3.11 | **2.83 ± 4.36** | **2.17 ± 3.19** | 3.61 ± 4.98 |
| Patient out of the room | **2.94 ± 4.40** | 0.61 ± 0.98 | 3.78 ± 4.94 | **0.50 ± 0.86** |

model relations between both modalities in a long-term sequence (Pérez et al., 2024) (Fig. 2.B). Finally, we use a classification head to assign event labels to the input frames.

## 3. Experimental Validation

**Dataset:** We assessed our method on an OR workflow dataset created using role play as part of the HALO (Hopkins Ambient Learning and Optimization) effort. The dataset comprises 38 simulated surgical trials across 7 ORs at Johns Hopkins Hospital. It includes videos and event annotations for 5 events and bounding box annotations for 14 objects. **Evaluation Metric:** We used mean Average Precision (mAP) at various temporal Intersection over Union (tIoU) thresholds to evaluate the performance of our event detection, a standard metric for Temporal Action Localization. Also, we computed the temporal prediction error between the predicted and ground truth (GT) start and end times for each event class. **Experiments:** We compared the performance of models trained on different input representations. As shown in Table 1, our Mask-Depth DT achieves the highest average mAP of 72.93%, outperforming both the RGB input baseline and single-modality approaches. Also, the reduced temporal prediction errors of Table 2 indicate precise event boundary detection using our DT-based approach. These results demonstrate that our DT effectively extracts geometric and semantic abstractions from the physical environment, creating a comprehensive anonymized representation of the OR that allows privacy-preserving workflow analysis.

## 4. Conclusions

DTs facilitate cross-institutional model training and improves model generalizability by reducing domain-specific visual appearances (Ding et al., 2024b). Our study demonstrates that these DTs enable accurate privacy-preserving OR workflow analysis and validate their potential application in OR modeling.

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
