# OpenReview forum: "Privacy-Preserving Operating Room Workflow Analysis using Digital Twins"
_MIDL.io/2025/Short_Papers — MIDL 2025 - Short Papers_

### Official Review · Reviewer_ebkL · 2025-04-23

**Rating:** 5
**Confidence:** 4

**Summary:**

The authors propose a two-stage pipeline for privacy-preserving operating room (OR) video analysis and event detection. First, they leverage a vision foundation model for depth estimation and semantic segmentation to generate de-identified digital twins of the OR from RGB videos. Second, they employ the SafeOR model to process the segmentation and depth maps for OR event detection. Their model is evaluated on an internal dataset of 38 simulated surgical trials with five event classes, demonstrating performance comparable to or better than that of raw RGB video-based detection.

**Strengths:**

- The paper is well written and structured.
- The related work discusses face blurring, light cameras, and their limitations.
- Results demonstrate performance comparable to or better than raw RGB video-based detection.

**Weaknesses:**

- The authors could consider discussing the limitations of their methods and potential directions for future work.

---

### Decision · Program_Chairs · 2025-05-01

Accept